# Mid-level Visual Element Discovery
# as Discriminative Mode Seeking

**Carl Doersch**
Carnegie Mellon University
cdoersch@cs.cmu.edu

**Abhinav Gupta**
Carnegie Mellon University
abhinavg@cs.cmu.edu

**Alexei A. Efros**
UC Berkeley
efros@cs.berkeley.edu

## Abstract

Recent work on mid-level visual representations aims to capture information at the level of complexity higher than typical "visual words", but lower than full-blown semantic objects. Several approaches [5, 6, 12, 23] have been proposed to discover *mid-level visual elements*, that are both 1) representative, *i.e.*, frequently occurring within a visual dataset, and 2) visually discriminative. However, the current approaches are rather *ad hoc* and difficult to analyze and evaluate. In this work, we pose visual element discovery as discriminative mode seeking, drawing connections to the the well-known and well-studied mean-shift algorithm [2, 1, 4, 8]. Given a weakly-labeled image collection, our method discovers visually-coherent patch clusters that are maximally discriminative with respect to the labels. One advantage of our formulation is that it requires only a single pass through the data. We also propose the Purity-Coverage plot as a principled way of experimentally analyzing and evaluating different visual discovery approaches, and compare our method against prior work on the Paris Street View dataset of [5]. We also evaluate our method on the task of scene classification, demonstrating state-of-the-art performance on the MIT Scene-67 dataset.

## 1   Introduction

In terms of sheer size, visual data is, by most accounts, the biggest "Big Data" out there. But, unfortunately, most machine learning algorithms (with some notable exceptions, e.g. [13]) are not equipped to handle it directly, at the raw pixel level, making research on finding good visual representations particularly relevant and timely. Currently, the most popular visual representations in machine learning are based on "visual words" [24], which are obtained by unsupervised clustering ($k$-means) of local features (SIFT) over a large dataset. However, "visual words" is a very low-level representation, mostly capturing local edges and corners ([21] notes that "visual letters" or "visual phonemes" would have been a more accurate term). Part of the problem is that the local SIFT features are relatively low-dimensional (128D), and might not be powerful enough to capture anything of higher complexity. However, switching to a more descriptive feature (*e.g.* $2,000$-dimensional HOG) causes k-means to produce visually poor clusters due to the curse of dimensionality [5].

Recently, several approaches [5, 6, 11, 12, 15, 23, 26, 27] have proposed mining visual data for discriminative *mid-level visual elements*, *i.e.*, entities which are more informative than "visual words," and more frequently occurring and easier to detect than high-level objects. Most such approaches require some form of weak per-image labels, *e.g.*, scene categories [12] or GPS coordinates [5] (but can also run unsupervised [23]), and have been recently used for tasks including image classification [12, 23, 27], object detection [6], visual data mining [5, 15], action recognition [11], and geometry estimation [7]. But how are informative visual elements to be identified in the weakly-labeled visual dataset? The idea is to search for clusters of image patches that are both 1) representative, *i.e.* frequently occurring within the dataset, and 2) visually discriminative. Unfortunately, algorithms for finding patches that fit these criteria remain rather ad-hoc and poorly understood. and often do not even directly optimize these criteria. Hence, our goal in this work is to quantify the terms "representative" and "discriminative," and show that a formulation which draws inspiration from

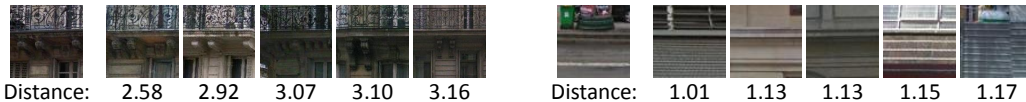

Distance: 2.58   2.92   3.07   3.10   3.16     Distance: 1.01   1.13   1.13   1.15   1.17

Figure 1: The distribution of patches in HOG feature space is very non-uniform and absolute distances cannot be trusted. We show two patches with their 5 nearest-neighbors from the Paris Street View dataset [5]; beneath each nearest neighbor is its distance from query. Although the nearest neighbors on the left are visually much better, their distances are more than twice those on the right, meaning that the actual densities of the two regions will differ by a factor of more than $2^d$, where $d$ is the intrinsic dimensionality of patch feature space. Since this is a 2112-dimensional feature space, we estimate $d$ to be on the order of hundreds.

the well-known, well-understood mean-shift algorithm can produce visual elements that are more representative and discriminative than those of previous approaches.

Mining visual elements from a large dataset is difficult for a number of reasons. First, the search space is huge: a typical dataset for visual data mining has tens of thousands of images, and finding something in an image (*e.g.*, finding matches for a visual template) involves searching across tens of thousands of patches at different positions and scales. To make matters worse, patch descriptors tend to be on the order of thousands of dimensions; not only is the curse of dimensionality a constant problem, but we must sift through terabytes of data. And we are searching for a needle in a haystack: the vast majority of patches are actually uninteresting, either because they are rare (*e.g.*, they may contain multiple random things in a configuration that never occurs again) or they are redundant due to the overlapping nature of patches. This suggests the need for an online algorithm, because we wish to discard much of the data while making as few passes through the dataset as possible.

The well-known mean-shift algorithm [2, 3, 8] has been proposed to address many of these problems. The goal of mean-shift is to find the local maxima (modes) of a density using a sample from that density. Intuitively, mean-shift initializes each cluster centroid to a single data point, then iteratively 1) finds data points that are sufficiently similar to each centroid, and, 2) averages these data points to update the cluster centroid. In the end, each cluster generally depends on only a tiny fraction of the data, thus eliminating the need to keep the entire dataset in memory.

However, there is one issue with using classical mean-shift to solve our problem directly: it only finds local maxima of a single, unlabeled density, which may not be discriminative. But in our case, we can use the weak labels to divide our data into two different subsets ("positive" (+) and "negative" (−)) and seek visual elements which appear only in the "positive" set and not in the "negative" set. That is, we want to find points in feature space where the density of the positive set is large, and the density of the negative set is small. This can be achieved by maximizing the well-studied density ratio $p_+(x)/p_-(x)$ instead of maximizing the density. While a number of algorithms exist for estimating ratios of densities (see [25] for a review), we did not find any that were particularly suitable for finding local maxima of density ratios. Hence, the first contribution of our paper is to propose a discriminative variant of mean-shift for finding visual elements. Similar to the way mean-shift performs gradient ascent on a density estimate, our algorithm performs gradient ascent on the density ratio (section 2). When we perform gradient ascent separately for each element as in standard mean-shift, however, we find that the most frequently-occuring elements tend to be over-represented. Hence, section 3 describes a modification to our gradient ascent algorithm which uses inter-element communication to approximate common adaptive bandwidth procedures. Finally, in section 4 we demonstrate that our algorithms produce visual elements which are more representative and discriminative than previous methods, and in section 5 we show they significantly improve performance in scene classification.

## 2   Mode Seeking on Density Ratios

Our goal is to extract discriminative visual elements by finding the local maxima of the density ratio. However, one issue with performing gradient ascent directly on standard density ratio estimates is that common estimators tend to use a fixed kernel bandwidth, for example:

$$\hat{r}(x) \propto \sum_{i=1}^{n} \theta_i K(\|x - x_i\|/h)$$

where $\hat{r}$ is the ratio estimate, the parameters $\theta_i \in \mathbb{R}$ are weights associated with each datapoint, $K$ is a kernel function (*e.g.*, a Gaussian), and $h$ is a globally-shared bandwidth parameter. The

bandwidth defines how much the density is smoothed before gradient ascent is performed, meaning these estimators assume a roughly equal distribution of points in all regions of the space. Unfortunately, absolute distances in HOG feature space cannot be trusted, as shown in Figure 1: any kernel bandwidth which is large enough to work well in the left example will be far too large to work well in the right. One way to deal with the non-uniformity of the feature space is to use an *adaptive* bandwidth [4]: that is, different bandwidths are used in different regions of the space. However, previous algorithms are difficult to implement for large data in high-dimensional spaces; [4], for instance, requires a density estimate for every point used in computing the gradient of their objective, because their formulation relies on a per-point bandwidth rather than a per-cluster bandwidth. In our case, this is prohibitively expensive. While approximations exist [9], they rely on approximate nearest neighbor algorithms, which work for low-dimensional spaces ($\leq 48$ dimensions in [9]), but empirically we have found poor performance in HOG feature space ($> 2000$ dimensions). Hence, we take a different approach which we have tailored for density ratios.

We begin by using a result from [2] that classical mean-shift (using a flat kernel) is equivalent to finding the local maxima of the following density estimate:

$$\frac{\sum_{i=1}^{n} \max(b - d(x_i, w), 0)}{z(b)} \tag{1}$$

In standard mean-shift, $d$ is the Euclidean distance function, $b$ is a constant that controls the kernel bandwidth, and $z(b)$ is a normalization constant. Here, the flat kernel has been replaced by its *shadow kernel*, the triangular kernel, using Theorem 1 from [2]. We want to maximize the density ratio, so we simply divide the two density estimates. We allow an adaptive bandwidth, but rather than associating a bandwidth with each datapoint, we compute it as a function of $w$ which depends on the data.

$$\frac{\sum_{i=1}^{n_{pos}} \max(B(w) - d(x_i^+, w), 0)}{\sum_{i=1}^{n_{neg}} \max(B(w) - d(x_i^-, w), 0)} \tag{2}$$

Where the normalization term $z(b)$ is cancelled. This expression, however, produces poor estimates of the ratio if the denominator is allowed to shrink to zero; in fact, it can produce arbitrarily large but spurious local maxima. Hence, we define $B(w)$ as the value of $b$ which satisfies:

$$\sum_{i=1}^{n_{neg}} \max(b - d(x_i^-, w), 0) = \beta \tag{3}$$

Where $\beta$ is a constant analogous to the bandwidth parameter, except that it directly controls how many negative datapoints are in each cluster. Note the value of the sum is strictly increasing in $b$ when it is nonzero, so the $b$ satisfying the constraint is unique. With this definition of $B(w)$, we are actually fixing the value of the denominator of (2) (We include the denominator here only to make the ratio explicit, and we will drop it in later formula). This approach makes the implicit assumption that the distribution of the negatives captures the overall density of the patch space. Note that if we assume the denominator distribution is uniform, then $B(w)$ becomes fixed and our objective is identical to fixed-bandwidth mean-shift.

Returning to our formulation, we must still choose the distance function $d$. In high-dimensional feature space, [20] suggests that normalized correlation provides a better metric than the Euclidean distance commonly used in mean-shift. Formulations of mean-shift exist for data constrained to the unit sphere [1], but again we must adapt them to the ratio setting. Surprisingly, replacing the Euclidean distance with normalized correlation leads to a simpler optimization problem. First, we mean-subtract and normalize all datapoints $x_i$ and rewrite (2) as:

$$\sum_{i=1}^{n_{pos}} \max(w^\top x_i^+ - b, 0) \quad \text{s.t.} \quad \begin{array}{c} \sum_{i=1}^{n_{neg}} \max(w^\top x_i^- - b, 0) = \beta \\ \|w\|^2 = 1 \end{array} \tag{4}$$

Where $B(w)$ has been replaced by $b$ as in equation (3), to emphasize that we can treat $B(w)$ as a constraint in an optimization problem. We can further rewrite the above equation as finding the local maxima of:

$$\sum_{i=1}^{n_{pos}} \max(w^\top x_i^+ - b, 0) - \lambda \|w\|^2 \quad \text{s.t.} \quad \sum_{i=1}^{n_{neg}} \max(w^\top x_i^- - b, 0) = \beta \tag{5}$$

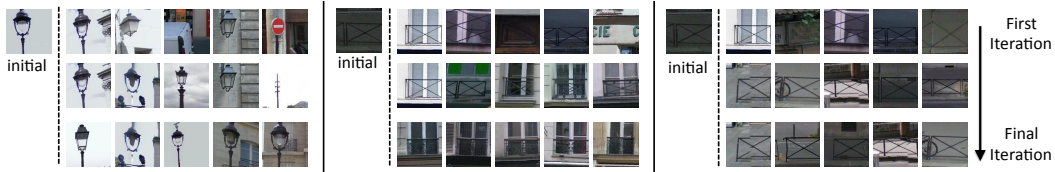

Figure 2: Left: without competition, the algorithm from section 2 correctly learns a street lamp element. Middle: The same algorithm trained on a sidewalk barrier, which is too similar to the very common "window with railing" element, which takes over the cluster. Right: with the algorithm from section 3, the window gets down-weighted and the algorithm can learn the sidewalk barrier.

Note that (5) is equivalent to (4) for some appropriate rescaling of $\lambda$ and $\beta$. It can be easily shown that multiplying $\lambda$ by a constant factor does not change the relative location of local maxima, as long as we divide $\beta$ by that same factor. Such a re-scaling will in fact result in re-scaling $w$ by the same value, so we can choose a $\lambda$ and $\beta$ which makes the norm of $w$ equal to 1. [1]

After this rewriting, we are left with an objective that looks curiously like a margin-based method. Indeed, the negative set is treated very much like the negative set in an SVM (we penalize the linear sum of the margin violations), which follows [23]. However, unlike [23], which makes the ad-hoc choice of 5 positive examples, our algorithm allows each cluster to select the optimal number of positives based on the decision boundary. This is somewhat reminiscent of unsupervised margin-based clustering [29, 16].

Mean-shift prescribes that we initialize the procedure outlined above at every datapoint. In our setting, however, this is not practical, so we instead use a randomly-sampled subset. We run this as an online algorithm by breaking the dataset into chunks and then mining, one chunk at a time, for patches where $w^\top x - b > -\epsilon$ for some small $\epsilon$, akin to "hard mining" for SVMs. We perform gradient ascent after each mining phase. An example result for this algorithm is shown in in Figure 2, and we include further results below. Gradient ascent on our objective is surprisingly efficient, as described in Appendix A.

## 3   Better Adaptive Bandwidth via Inter-Element Communication

Implicit in our formulation thus far is the idea that we do not want a single mode, but instead many distinct modes which each corresponds to a different element. In theory, mode-seeking will find every mode that is supported by the data. In practice, clusters often drift from weak modes to stronger modes, as demonstrated in Figure 2 (middle). One way to deal with this is to assign smaller bandwidths to patches in dense regions of the space [4], *e.g.*, the window railing on row 1 of Figure 2 (middle) would hopefully have a smaller bandwidth and hence not match to the sidewalk barrier. However, estimating a bandwidth for every datapoint in our setting is not practical, so we seek an approach which only requires one pass through the data. Since patches in regions of the feature space with high density ratio will be members of many clusters, we want a mechanism that will reduce their bandwidth. To accomplish this, we extend the standard local (per-element) optimization of mean-shift into a joint optimization among the $m$ different element clusters. Specifically, we control how a single patch can contribute to multiple clusters by introducing a *sharing weight* $\alpha_{i,j}$ for each patch $i$ that is contained in a cluster $j$, akin to soft-assignment in EM GMM fitting. Returning to our fomulation, we maximize (again with respect to the $w$'s and $b$'s):

$$\sum_{i=1}^{n_{pos}} \sum_{j=1}^{m} \alpha_{i,j} \max(w_j^\top x_i^+ - b_j, 0) - \lambda \sum_{j=1}^{m} \|w_j\|^2 \ \text{ s.t. } \forall j \ \sum_{i=1}^{n_{neg}} \max(w_j^\top x_i^- - b_j, 0) = \beta \quad (6)$$

Where each $\alpha_{i,j}$ is chosen such that any patch which is a member of multiple clusters gets a lower weight. (6) also has a natural interpretation in terms of maximizing the "representative-ness" of the set of clusters: clusters are rewarded for representing patches that are not repre-sented by other clusters. But how can we set the $\alpha$'s? One way is to set $\alpha_{i,j} = \max(w_j^\top x_i^+ - b_j, 0)/\sum_{k=1}^{m} \max(w_k^\top x_i^+ - b_k, 0)$, and alternate between setting the $\alpha$'s and optimizing the $w$'s and

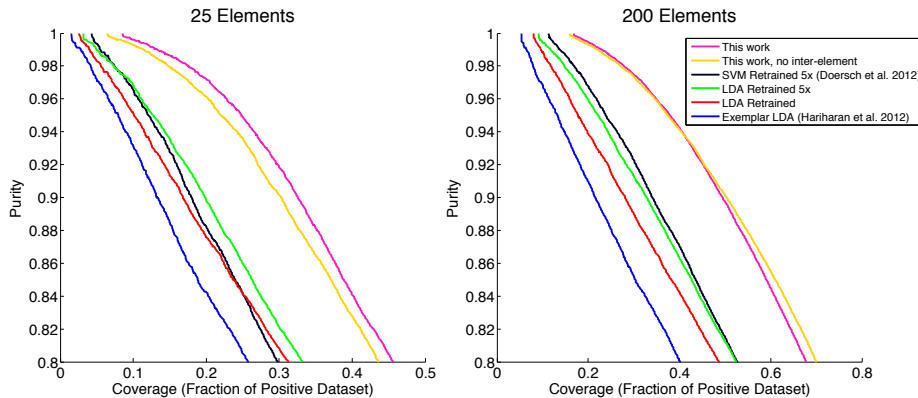

Figure 3: Purity-coverage graph for our algorithm and baselines. In each plot, purity measures the accuracy of the element detectors, whereas coverage captures how often they fire. Curves are computed over the top 25 (left) and 200 (right) elements. Higher is better.

$b$'s at each iteration. Intuitively, this algorithm would be much like EM, alternating between softly assigning cluster memberships for each datapoint and then optimizing each cluster. However, this goes against our mean-shift intuition: if two patches are really instances of the same element, then clusters initialized from those two points should converge to the same mode and not "compete" with one another. So, our heuristic is to first cluster the elements. Let $C_j$ be the assigned cluster for the $j$'th element. Then we set

$$\alpha_{i,j} = \frac{\max(w_j^\top x_i^+ - b_j, 0)}{\max(w_j^\top x_i^+ - b_j, 0) + \sum_{k=1}^m I(C_k \neq C_j) \max(w_k^\top x_i^+ - b_k, 0)} \quad (7)$$

In this way, any "competition" from elements that are too similar to each other is ignored. To obtain the clusters, we perform agglomerative (UPGMA) clustering on the set of element clusters, using the negative of the number of overlapping cluster members as a "distance" metric.

In practice, however, it is extremely rare that the exact same patch is a member of two different clusters; instead, clusters will have member patches that merely overlap with each other. Our heuristic deal with this is to compute a quantity $\alpha'_{i,j,p}$ which is analogous to the $\alpha_{i,j}$ defined above, but is defined for every pixel $p$. Then we compute $\alpha_{i,j}$ for a given patch by averaging $\alpha'_{i,j,p}$ over all pixels in the patch. Specifically, we compute $\alpha_{i,j}$ for patch $i$ as the mean over all pixels $p$ in that patch of the following quantity:

$$\alpha'_{i,j,p} = \frac{\max(w_j^\top x_i^+ - b_j, 0)}{\max(w_j^\top x_i^+ - b_j, 0) + \sum_{x \in Ov(p)} \sum_{k=1}^m I(C_k \neq C_j) \max(w_k^\top x_i^+ - b_k, 0)} \quad (8)$$

Where $Ov(p)$ denotes the set of features for positive patches that contain the pixel $p$.

It is admittedly difficult to analyze how well these heuristics approximate the adaptive bandwidth approach of [4], and even there the setting of the bandwidth for each datapoint has heuristic aspects. However, empirically our approach leads to improvements in performance as discussed below, and suggests a potential area for future work.

## 4    Evaluation via Purity-Coverage Plot

Our aim is to discover visual elements that are maximally representative and discriminative. To measure this, we define two quantities for a set of visual elements: **coverage** (which captures representativeness) and **purity** (which captures discriminativeness). Given a held-out test set, visual elements will generate a set of patch detections. We define the coverage of this set of patches to be the fraction of the pixels from the positive images claimed by at least one patch. We define the purity of a set as the percentage of the patches that share the same label. For an individual visual element, of course, there is an inherent trade-off between purity and coverage: if we lower the detection threshold, we cover more pixels but also increase the likelihood of making mistakes. Hence, we can construct a purity-coverage curve for a set of elements, analogous to a precision-recall curve. We could perform this analysis on any dataset containing positive and negative images, but [5] presents a dataset which is particularly suitable. The goal is to mine visual elements which define the look and feel of a geographical locale, with a training set of 2,000 Paris Street View images and 8,000

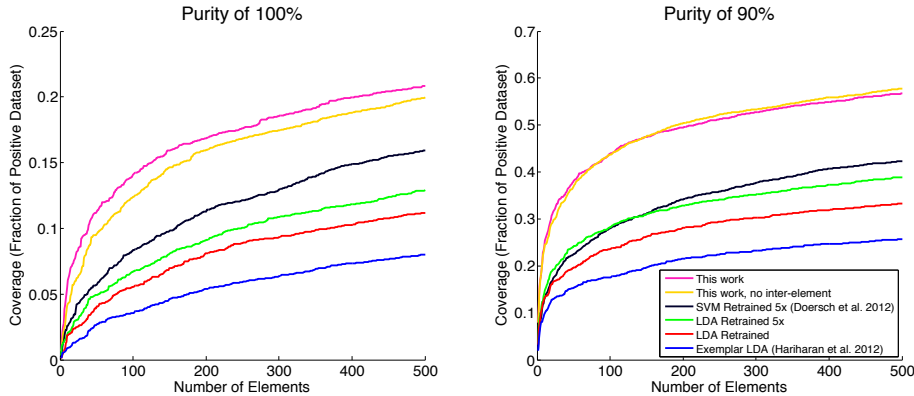

Figure 4: Coverage versus the number of elements used in the representation. On the left we keep only the detections with a score higher than the score of the detector's first error (*i.e.* purity 1). On the right, we lower the detection threshold until the elements are 90% pure. Note: this is the same purity and coverage measure for the same elements as Figure 3, just plotted differently.

non-Paris images, as well as 2,999 of both classes for testing. Purity-coverage curves for this dataset are shown in Figure 3.

To plot the curve for a given value of purity $p$, we rank all patches by $w^\top x - b$ independently for every element, and select, for a given element, all patches up until the last point where the element has the desired purity. We then compute the coverage as the union of patches selected for every element. Because we are taking a union of patches, adding more elements can only increase coverage, but in practice we prefer concise representations, both for interpretability and for computational reasons. Hence, to compare two element discovery methods, we must select exactly the same number of elements for both of them. Different works have proposed different heuristics for selecting elements, which would make the resulting curves incomparable. Hence, we select elements in the same way for all algorithms, which approximates an "ideal" selection for our measure. Specifically, we first fix a level of purity (95%) and greedily select elements to maximize coverage (on the testing data) for that level of purity. Hence, this ranking serves as an oracle to choose the "best" set of elements for covering the dataset at that level of purity. While this ranking has a bias toward large elements (which inherently cover more pixels per detection), we believe that it provides a valuable comparison between algorithms. Our purity-coverage curves are shown in Figure 3, for the 25 and 200 top elements, respectively. We can also slice the same data differently, fixing a level of purity for all elements and varying the number of elements, as shown in Figure 4.

**Baselines:** We included five baselines of increasing complexity. Our goal is not only to analyze our own algorithm; we want to show the importance of the various components of previous algorithms as well. We initially train $20,000$ visual elements for all the baselines, and select the top elements using the method above. The simplest baseline is "Exemplar LDA," proposed by [10]. Each cluster is represented by a hyperplane which maximally separates a single seed patch from the negative dataset learned via LDA, *i.e.* the negative distribution is approximated using a single multivariate Gaussian. To show the effects of re-clustering, "LDA Retrained" takes the top 5 positive-set patches retrieved in Exemplar LDA (including the initial patch itself), and repeats LDA, separating those 5 from the negative Gaussian. This is much like the well-established method of "query expansion" for retrieval, and is similar to [12] (although they use multiple iterations of query expansion). Finally, "LDA Retrained 5 times" begins with elements initialized via the LDA retraining method, and re-trains the LDA classifier, each time throwing out the previous top 5 used to train the previous LDA, and selecting a new top 5 from held-out data. This is much like the iterative SVM training of [5], except that it uses LDA instead of an SVM. Finally, we include the algorithm of [5], which is a weakly supervised version of [23], except that knn is being used for initialization instead of kmeans. The iterations of retraining clearly improve performance, and it seems that replacing LDA with an SVM also gives improvement, especially for difficult elements.

**Implementation details**: We use the same patch descriptors described in [5] and whiten them following [10]. We mine elements using the online version of our algorithm, with a chunk size of 1000 (200 Paris, 800 non-Paris per batch). We set $\beta * \lambda = t/500$ where $t$ is the iteration number, such that the bandwidth increases proportional to the number of samples. We train the elements for about 200

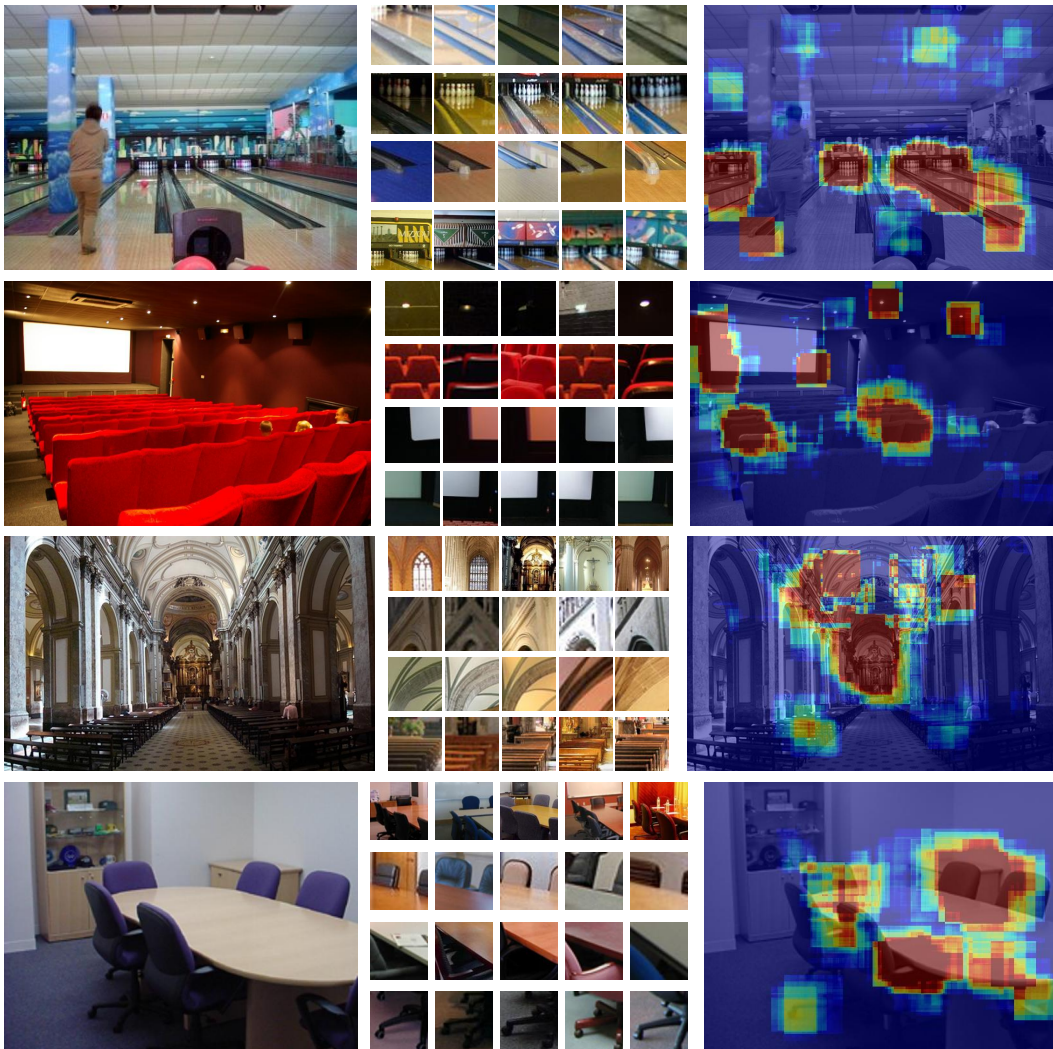

Figure 5: For each correctly classified image (left), we show four elements (center) and heatmap of the locations (right) that contributed most to the classification.

Table 1: Results on MIT 67 scenes

| | | | | | |
|---|---|---|---|---|---|
| ROI + Gist [19] | 26.05 | D-Patches [23] | 38.10 | D-Parts [26] | 51.40 |
| MM-scene [30] | 28.00 | LPR [22] | 44.84 | IFV [12] | 60.77 |
| DPM [17] | 30.40 | BoP [12] | 46.10 | BoP+IFV [12] | 63.10 |
| CENTRIST [28] | 36.90 | miSVM [15] | 46.40 | Ours (no inter-element, §2) | **63.36** |
| Object Bank [14] | 37.60 | D-Patches (full) [23] | 49.40 | Ours (§3) | **64.03** |
| RBoW [18] | 37.93 | MMDL [27] | 50.15 | Ours+IFV | **66.87** |

gradient steps after each chunk of mining. To compute $\alpha_{i,j}$ for patch $i$ and detector $j$, we actually use scale-space voxels rather than pixels, since a large detection can completely cover a small detection but not vice versa. Hence, the set of scale-space voxels covered is a 3D box, the width of the bounding box by its height (both discretized by a factor of 8 for efficiency) by 5, covering exactly one "octave" of scale space (*i.e.* $log2(\sqrt{width * height}) * 5$ through $log2(\sqrt{width * height}) * 5 + 4$). For experiments without inter-element communication, we simply set $\alpha_{i,j}$ to .1. Finally, to reduce the impact of highly redundant textures, we divide $\alpha_{i,j}$ divided by the total number of detections for element $j$ in the image containing $i$. Source code will be available online.

## 5   Scene Classification

Finally, we evaluate whether our visual element representation is useful for scene classification. We use the MIT Scene-67 dataset [19], where machine performance remains substantially below human

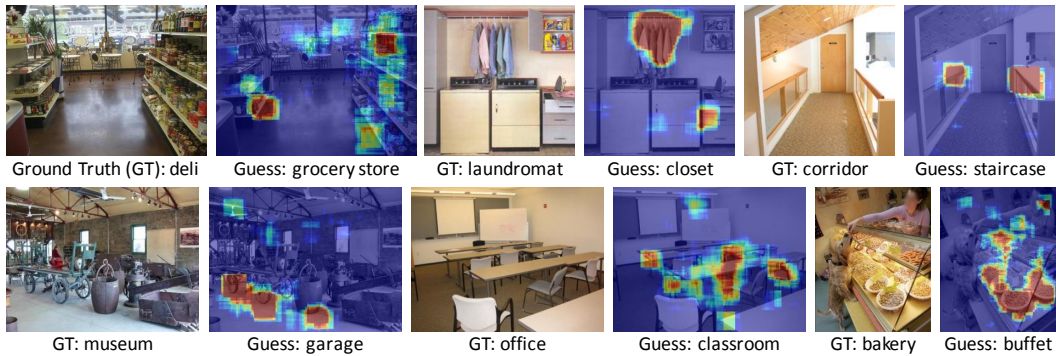

| Ground Truth (GT): deli | Guess: grocery store | GT: laundromat | Guess: closet | GT: corridor | Guess: staircase |
| GT: museum | Guess: garage | GT: office | Guess: classroom | GT: bakery | Guess: buffet |

Figure 6: Each of these images was misclassified by the algorithm, and the heatmaps explain why. For instance, it may not be obvious why a corridor would be classified as a staircase, but we can see (top right) that the algorithm has identified the railings as a key staircase element, and has found no other staircase elements the image.

performance. For indoor scenes, objects within the scene are often more useful features than global scene statistics [12]: for instance, shoe shops are similar to other stores in global layout, but they mostly contain shoes.

**Implementation details**: We used the original Indoor-67 train/test splits (80 training and 20 testing images per class). We learned 1600 elements per class, for a total of $107,200$ elements, following the procedure described above. We include right-left flipped images as extra positives. 5 batches were sufficient, as this dataset is smaller. We also used smaller descriptors: 6-by-6 HOG cells, corresponding to 64-by-64 patches and 1188-dimensional descriptors. We again select elements by fixing purity and greedily selecting elements to maximize coverage, as above. However, rather than defining coverage as the number of pixels (which is biased toward larger elements), we simply count the detections, penalizing for overlap: we penalize each individual detection by a factor of $1/(1 + n_{overlap})$, where $n_{overlap}$ is the number of detections from previously selected detectors that a given detection overlaps with. We select 200 top elements per class. To construct our final feature vector, we use a 2-level (1x1 and 2x2) spatial pyramid and take the max score per detector per region, thresholded at $-.5$ (since below this value we do not expect the detection scores to be meaningful) resulting in a 67,000-dimensional vector. We average the feature vector for the right and left flips of the image, and classify using 67 one-vs-all linear SVM's. Note that this differs from [23], which selects only the elements for a given class in each class-specific SVM.

Figure 5 shows a few qualitative results of our algorithm. Quantitative results and comparisons are shown in Table 1. We significantly outperform other methods based on discriminative patches, suggesting that our training method is useful. We even outperform the Improved Fisher Vector of [12], as well as IFV combined with discriminative patches (IFV+BoP). Finally, although the optimally-performing representation is dense (about 58% of features are nonzero), it can be made much sparser without sacrificing much performance. For instance, if we trivially zero-out low-valued features until fewer than 6% are nonzero, we still achieve 60.45% accuracy.

# 6 Conclusion

We developed an extension of the classic mean-shift algorithm to density ratio estimation, showing that the resulting algorithm could be used for element discovery, and demonstrating state-of-the-art results for scene classification. However, there is still much room for improvement in weakly-supervised element discovery algorithms. For instance, our algorithm is limited to binary labels, but image labels may be continuous (*e.g.*, GPS coordinates or dates). Also, our elements are detected based only on individual patches, but images often contain global structures beyond patches.

**Acknowledgements:** We thank Abhinav Shrivastava, Yong Jae Lee, Supreeth Achar, and Geoff Gordon for helpful insights and discussions. This work was partially supported by NDSEG fellowship to CD, An Amazon Web Services grant, a Google Research grant, ONR MURI N000141010934, and IARPA via Air Force Research Laboratory. The U.S. Government is authorized to reproduce and distribute reprints for governmental purposes notwithstanding any copyright annotation thereon. Disclaimer: The views and conclusions contained herein are those of the authors and should not be interpreted as necessarily representing the official policies or endorsements, either expressed or implied, of IARPA, AFRL or the U.S. Government.

## Footnotes

[1]Admittedly this means that the norm of $w$ has an indirect effect on the underlying bandwidth: specifically if the norm of $w$ is increased, it has a similar effect as a proportional derease in $\beta$ in (4). However, since $w$ is roughly proportional to the density of the *positive* data, the bandwidth is only reduced when the density of positive data is high.

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
