[Supplementary Material · nips2013_appendix.pdf]

# A  Optimizing the objective

Algorithm 1 gives a summary of our optimization procedure. We begin by sampling a set of patches from the positive dataset, and initialize our $w_j$ vectors as the features for these patches. We initialize $b_j$ to 0. For simplicity of notation in this section, we append $b_j$ to $w_j$ and append a $-1$ to each feature vector $x$. We can then "mine" through a set of images for patches where $w_j^\top x > 0$ for some $j$. In practice, it greatly improves computational efficiency to have a separate round of mining initially on a small set of negative images, where we only update $b_j$ to satisfy the constraint of (6).

After a round of mining on a single chunk of the data (including positives and negatives), we set the $\alpha$'s according to the procedure described in section 3. We must then optimize the following:

$$\sum_{i=1}^{n_{pos}} \alpha_{i,j} \max(w_j^\top x_i^+, 0) - \lambda \sum_{j=1}^{m} \|[w_j]_{1:d}\|^2 \quad \text{s.t.} \quad \sum_{i=1}^{n_{neg}} \max(w_j^\top x_i^-, 0) \leq \beta \tag{9}$$

Here, $d$ is the data dimensionality, and $[\cdot]_{1:d}$ selects the first $d$ components of the vector such that the bias term is excluded. Note that we can replace the $=$ with a $\leq$ in the constraint because it does not affect the solution: a decrease in $b$ will always increase the objective, and hence the inequality constraint will always be tight at the solution. With this modification, it is straightforward to show that the constraint defines a convex set. At first glance, Expression (9) seems quite difficult to optimize, as we are maximizing a non-concave function. It is unlikely that a convex relaxation will be useful either, because different elements correspond to different local maxima of the objective. In practice, however, we can approximately optimize (9) directly, and do so efficiently. First, note that *locally* the function is a simple quadratic on an affine subspace, as long as $w_j$ remains in a neighborhood where the sign of $w_j^\top x$ does not change for any $x$. Hence, we perform a form of projected gradient descent; pseudocode is given in the `optimize` function of Algorithm 1. We first compute the gradient of (9) and then find its projection $\nabla$ onto the current affine subspace, *i.e.*, the space defined by:

$$\nabla^\top \sum_{i=1}^{n_{neg}} x_i^- I(w_j^\top x_i^- > 0) = 0 \tag{10}$$

where $I$ is the indicator function. This means that small updates in the direction $\nabla$ will not result in constraint violations. Next, we perform a line search on $w + t\nabla$, where $t$ is the step size that we search over:

$$t^* = \arg\max_t \sum_{i=1}^{n_{pos}} \alpha_{i,j}(w_j + t\nabla)^\top x_i^+ * I(w_j^\top x_i^+ \geq 0) - \lambda\|[w_j + t\nabla]_{1:d}\|^2 \tag{11}$$

This is a simple quadratic that can be solved analytically. If the maximum $t^*$ of the line search does not cause $w_j^\top x$ to change for any $x$, then we accept this maximum, set $w_j = w_j + t^*\nabla$, and iterate. Otherwise, we set $t$ equal to a pre-determined fixed constant, and update. If the step causes $w_j^\top x_i^-$ to change sign for some $x_i^-$, however, then we will no longer satisfy the constraint in (9). Ideally, we would orthogonally project $w_j$ onto the constraint set, but finding the correct orthogonal projection is computationally expensive. Hence, we approximate the projection operator with gradient descent (with respect to $w_j$) on the expression:

$$\left| \sum_{i=1}^{n_{neg}} \max(w_j^\top x_i^-, 0) - \beta \right| \tag{12}$$

This procedure is shown in the `satisfyConstrains` function of Algorithm 1. This function is piecewise linear, so gradient descent can be performed very efficiently. If the path of gradient descent is a straight line (*i.e.* for no $x$ does $w^\top x$ change sign) then this will be a proper projection, but otherwise it is an approximation. In practice we run the optimization on a fixed computational budget for each element, since in practice we find that learning more elements is more useful than optimizing individual elements more exactly.

**Algorithm 1**: Discriminative Mode Seeking Pseudocode

---

**Data**: $I^+, I^-$: positive and negative image sets

Initialize $W = [w_1, ..., w_m]$ as random patches from positive images, with the last (bias) row 0

Initialize $B = [b_1, ..., b_m]$ by running W on a subset of $I^-$ and finding $b$'s that satisfy 3

Set the last row of $W$ equal to $B$.

Distribute $I^+$ and $I^-$ evenly into $l$ sets, $I_1, ..., I_L$

**for** $l \leftarrow 1$ **to** $L$ **do**

    Mine for patches $x$ in $I_l$ for which any of $W^\top x > 0$

    **for** $j \leftarrow 1$ **to** $m$ **do**

        $X \leftarrow$ the set of $x$ for which $w_j^\top x > 0$

        $[w_j] \leftarrow$ `optimize`$(w_j, X)$

    **end**

**end**

---

**Function** `optimize`$(w, X)$

---

$X^+, X^- \leftarrow$ Positive and negative examples from X, respectively;

**while** *not converged and not timed out* **do**

    $\triangledown \leftarrow \sum_{x \in X^+, w^\top x > 0} x - 2 * \lambda \|[w]_{1:d}\| x$ ;        `// Gradient of objective`

    $\Pi \leftarrow \sum_{x \in X^-, w^\top x > 0} x$;                `// Gradient of constraint`

    $\triangledown \leftarrow (\Pi \triangledown^\top \Pi)/\|\Pi\|^2$;            `// Project` $\triangledown$ `to be orthogonal to` $\Pi$

    $w \leftarrow w + t * \triangledown$;                `// take a step of size` $t$ `(see text)`

    $w \leftarrow$ satisfyConstraints$(w, X^-)$;

**end**

**return** w;

---

**Function** `satisfyConstraints`$(w, X^-)$

---

**while** *constraint is not satisfied* **do**

    $\Pi \leftarrow$ sum of $x \in X^-$ where $w^\top x > 0$;        `// Gradient of constraint`

    $\delta \leftarrow \min \delta$ such that the sign of $(w - \delta * \Pi)^\top x$ changes for some $x \in X^-$;

    **if** *some $\delta_0 < \delta$ makes $(w - \delta_0 * \Pi)$ satisfy the constraint* **then**

        $\delta \leftarrow \delta_0$;

    **end**

    $w \leftarrow w - \delta * \Pi$;

**end**

**return** w;