[Reviews · NeurIPS 2013]

Submitted by Assigned_Reviewer_7

This paper tackles the problem of discovering mid-level features using discriminative mean-shift. The algorithm proposed by the authors involves the modification of the standard mean-shift algorithm to a discriminative setting such that it can be applied to discover mid-level patches. The authors also propose a more principled way of evaluating the results of mid-level patch discovery using purity-coverage curves, and show that their algorithm outperforms existing methods in these metrics. In addition, their algorithm significantly outperforms existing methods on the commonly used MIT 67-scene dataset when used without fisher vectors, and is improved slightly when combined. While the overall performance improvement as compared to BoP+IFV [8] is "only" 2.7% (still significant), the performance when using mid-level patches alone improves significantly as compared to BoP [8] alone i.e. 46% vs 64%.

Overall, I feel that the use of the discriminative mean-shift algorithm presented is novel, especially in this detection like setting, and the results are very promising. The paper is well written and the results are well analyzed. Thus, I would recommend that this paper be accepted to NIPS.

Some minor concerns/suggestions:
- What is the running time of your algorithm, and how does it compare to existing methods?
- Will you release the source code after publication?
- It might be worthwhile citing this paper since it shares a common name and mentioning why it's different:
Wang, Junqiu, and Yasushi Yagi. "Discriminative mean shift tracking with auxiliary particles." Computer Vision–ACCV 2007. Springer Berlin Heidelberg, 2007. 576-585.
- It would be interesting to try this approach on SUN where object segmentation is available to identify object categories that are commonly selected by the mid-level patches. There is likely to be some semantic structure in the automatic discovery.
Summary: The paper proposes the idea of discriminative mean-shift for the discovery of mid-level visual elements. The paper has very promising results and is well written/analyzed.

Submitted by Assigned_Reviewer_10

This paper proposes a discriminative clustering algorithm inspired by mean shift and the idea of finding local maxima of the density ratio (ratio of the densities of positive and negative points). The work is motivated by recent approaches of [4,8,16] aimed at discovering distinctive mid-level parts or patches for various recognition tasks. In the authors' own words from the Intro, "The idea is to search for clusters of image patches that are both 1) representative... and 2) visually discriminative. Unfortunately, finding patches that fit these criteria remain rather ad-hoc and poorly understood. While most current algorithms use a discriminative clustering-like procedure, they generally don't optimize elements for these criteria, or do so in an indirect, procedural way that is difficult to analyze. Hence, our goal in this work is to quantify the terms 'representative' and 'discriminative', and show that that a generalization of the well-known, well-understood Mean-Shift algorithm can produce visual elements that are more representative and discriminative than those of previous approaches." I find this motivation very compelling and like the formulation of discriminative clustering in terms of maximizing the density ratio.

The derivation of the clustering objective starts out promisingly enough on p. 2-3, by p. 4 it makes a number of confusing leaps that seem to take it pretty far from the original formulation. Specifically, on lines 188-190, the authors comment on eq. (5), "representing a patch multiple times is treated the same as representing different patches. Ideally, none of the patches should be double-counted." This leads to the introduction of "sharing coefficients" that ensure competition between clusters. However, isn't "double-counting" actually necessary in order to accurately estimate the local density of the patches, i.e., the more similar patches there are close to a given location in the feature space, the higher the density should be? Or does "double-counting" refer to something else? This needs to be clarified. Next, the discussion following eq. (6) appears to introduce heuristic criteria for setting the sharing coefficients, and even more heuristics are needed to deal with overlapping patches (lines 199-211). As a result, by the time we get to the final objective (eq. 7), the original one (eq. 1) seems to have been abandoned or changed beyond recognition. While the authors start out rightly decrying the heuristic nature of approaches such as [4,8,16], they end up deriving something no less heuristic.

Strengths
=========

+ The idea of deriving a discriminative clustering algorithm by maximizing the density ratio is novel and compelling.

+ The experimental results are the main point in favor of accepting the paper. According to Table 1, the proposed approach outperforms even the very recent results of [8] on the MIT Indoor Scene dataset (however, see below).

Weaknesses
==========

- The derivation of the clustering objective makes several poorly explained leaps (see above). The authors need to better motivate the different steps of their derivation and explain the relationship to related methods. For one, while the paper is titled "Discriminative Mean Shift", the connection of the proposed method and mean shift is less than apparent: the original mean shift formulation is purely local (each point in the feature space finds its closest local maximum), while the method derived in this paper appears to globally optimize over cluster centers by introducing a "competition" criterion. If I understand correctly, this is not simply a difference of maximizing local density vs. density ratio. A better discussion would be helpful. Also, the final objective (eq. 7) has a strong margin-based feel to it. Similarity (or lack thereof) to other existing margin-based clustering approaches should be discussed.

- For that matter, there should be citations to other existing discriminative clustering formulations, such as

Linli Xu, James Neufeld, Bryce Larson, and Dale Schuurmans, Maximum margin clustering, NIPS 2005.

- The proposed optimization algorithm does not appear to have any theoretical guarantees (lines 257-259).

- While the experimental results appear extremely strong, it is hard to get any insight as to why the proposed method outperforms very recent well-engineered systems such as those of [8]. Is this due to the clustering objective or to other implementation details described in lines 406-422? Since there are many potential implementation differences between the proposed approach and the baselines it compares against (including feature extraction, pre- and post-processing, classification), the causes behind the superior performance reported in this paper are not at all clear. Some of the claims that are made in the experimental section are not supported by any quantitative or qualitative evidence shown in the paper, e.g., "small objects within the scene are often more useful features than global scene statistics: for instance, shoe shops are similar to other stores in global layout, but they mostly contain shoes."

Summary: This paper is above the acceptance threshold owing primarily to the strong experimental results, but the derivation of the clustering method is not clearly presented and appears to have several poorly motivated heuristic steps.

Submitted by Assigned_Reviewer_11

The paper proposes a new method for learning discriminative image patches predictive of the image class. The procedure starts by considering all (?) patches in an image collection. The discriminative patches are then found as the centres of patch clusters, obtained by a discriminative version of the mean-shift algorithm. Discriminativeness is incorporated in mean shift by dividing the kernel density estimate of the positive patches by the one of negative ones.

Pros:

- The problem of learning discriminative patches/parts is an important one.
- The classification performance of the proposed discriminative patches is very good, at least on MIT Scene 67.
- There are a few interesting insights in the formulation.

Cons:

- The authors start from mean-shift and gradually transform it into a very different algorithm. In the end, I am not sure that the mean-shift interpretation is useful at all.
- Several aspects of the formulation and derivation are heuristics. Some of the heuristics are changed on a dataset-by-dataset basis.
- The learning algorithm is also heuristic, with no formal guarantee of correctness/convergence.
- There are several small errors in the formal derivation.

Detailed comments:

The empirical results are sufficiently strong that this paper should be considered for publication. However, the formulation and technical derivation should be improved significantly, as detailed below:


### l.125: The notation should be clarified. The expression max(d(x_i,w) - b, 0) is the triangular kernel assuming that d() is the negative of the Euclidean distance (as stated on l. 130) *and* that the bandwidth b is negative, which is counter-intuitive. max(b - d(x_i,w), 0) is more natural.


### Eq. (1). arglocalmax is never defined.


### Eq. (2). This equation indicates that, contrary to what stated in the manuscript, the algorithm is not maximizing a ratio of density values, but an energy function computed with a sort of adaptive bandwidth. This density is given by

E(w) = sum_i max(d(x_i^+) - b(w), 0)

where the bandwidth b(w) is selected as a function of the current point w as

b(w) = b : sum_i max(d(x_i^+) - b, 0) = epsilon.

Several aspects of this formulation should be clarified:

(a) The triangular kernel should be normalized by its mass to get a proper density estimator. Interestingly, this normalization factor, which depends on w, cancels out in the ratio (2), which perhaps "saves the day".

(b) This formulation should be contrasted to the standard adaptive mean shift (e.g. B. Georgescu, I. Shimshoni, and P. Meer. Mean shift based clustering in high dimensions: A texture classification example. In Proc. ICCV, 2003). There the bandwidth is chosen as a function of x_i rather than w and the normalization of the kernels become crucial.

(c) What happens if there are more than one value of b(w) satisfying the constraint in (2) ?


### Eq. (3)

l. 157: It is the _squared_ Euclidean distance d^2() that reduces to the inner product, not the euclidean distance d().

Note that restricting the domain to the unit sphere requires in principle to modify all the densities to have this manifold as domain. Fortunately, the required modification (normalising factors) does not seem to have a consequence in this case.

l. 177: it seems to me that changing lambda _does_ change the solution w, not just its norm. To keep the direction of w invariant while changing lambda, epsilon must change as well. Therefore, choosing different values of lambda should have an effect on the solution, unless all values of epsilon are equally good (but then why having epsilon in the first place?).


### Eq. (5)

This is where the proposed method diverges substantially from mean shift. Mean shift applies hill climbing to each w independently starting from w = x_i for all data points, in order to determine a clustering of the data x_i themselves. Here, instead, the authors formulate (5) and (6) as a method to "explain" all the data. Practical differences include:

- mean-shift is non-parametric, in the sense that the number of clusters is not specified a priori. Here the authors start with a fixed number of cluster centers w1...wK and optimise those to fit the data, which is more similar to K-means.

- the authors worry about the fact that "patches should not be double counted" and introduce a set of soft associations data-cluster alpha_ij. This is difficult to map in the standard semantic of mean-shift clustering, where the association of a data point x_i to a mode wk is obtained implicitly by the locality of the kernel estimator only. As the authors argue, alpha_ij establish a "competition" between modes to explain data points, which again is more similar to k-means.

- the way the alpha_ij are updated has little to do with the optimization of (6) and is completely ad hoc (l.199 - 211)


### Optimization method

Unfortunately this method is just an heuristic (l. 212-259).


### Experiments

Baselines:

[5,8] have mechanism to avoid or remove redundant patches, which do not seem to be incorporated in this baseline. Removing such redundant patches might affect Fig. 3, 4.

Scene-67 experiments: There are several tuning of the representation (e.g. number of HOG cells in a descriptor) that probably helps the method achieve state of the art results. While this is ok, the authors should consider re-running this experiment with the baseline discriminative patches obtained as in Fig. 4.


### Other minor comments

l.315: I have seen [5] and [8] in CVPR 2013 and it seems to me that they both have LDA retraining.
Summary: The problem of learning discriminative parts of visual classes is important and the results in this paper are very good. However, there are several minor technical problems in the derivation of the algorithm.

POST REBUTTAL COMMENTS

As noted by R10 and I, the derivation makes several unclear leaps. In fact, there are several minor formal errors in the paper that were highlighted in the reviews, none of which is addressed in the authors' rebuttal. The fact that several aspects of the method are heuristics, and such heuristics are tuned on a dataset basis, was not addressed either.

All reviewers agree to accept the paper on the ground of the solid experimental results; however, the AC may want to take into account the existence of these formal problems before reaching a decision.
Author Feedback

Author rebuttal: We would like to thank the reviewers for their careful reading of the paper, insightful comments, and helpful suggestions. Particularly in regards to the clarity of the presentation, we now see what the confusing points are and will use the reviewers’ suggestions to clean up the story (as well as fix some notational bugs and missed citations).

Overall, reviewers seem positive about our idea, results and analysis; but R10&11 voice concerns about some heuristic aspects of the algorithm, particularly after “competition” is added, line 181. We agree that this aspect of the formulation is not as simple or elegant as one might hope for. Perhaps we should have emphasized more strongly that the simpler algorithm, without competition, actually accounts for the majority of the performance gains we see. The yellow line in Figs. 3 and 4 is pure discriminative mean-shift from equation (5), where each element is optimized locally. Furthermore, we tried the experiment R11 suggests, running the algorithm using only the mean-shift formulation, without competition, on Indoor-67. The result was 61.64%, a loss of 2.5% from the full algorithm with competition, but still far outperforming other patch-based methods. In the revision, we plan to cleanly separate (into two sections) the principled, discriminative mean-shift formulation and the “competition heuristic”, and discuss the pros/cons of the latter.

Specific answers:

R10: Some of the claims that are made in the experimental section are not supported
These claims have been made based on studies in previous papers. The example given by R10 is rephrasing of the claim in [8]: “scenes are ....characterized by their global layout (e.g. corridor) and also those that are best characterized by the objects they contain (e.g. bookshop)”. We will put the citations to these claims.

R11: What happens if there are more than value of b(w) satisfying the constraint in (2)
\sum_{x} max(w^T x -b,0) will be strictly decreasing in b except when b is so large that the entire sum is 0; hence the b which satisfies the constraint is guaranteed to be unique for fixed w and epsilon>0.

R11: (line 177) it seems to me that changing lambda _does_ change the solution w...epsilon must change as well
Correct; epsilon must be scaled by the inverse of lambda. We will clarify this in the final version.

R7: running time?
Around 2000 cpu-hours for both experiments; i.e. it’s comparable to [4], although the current Matlab implementation could likely be sped up substantially.

R7: Source code release?
Yes, we are committed to releasing all the source code and results.

R11: [5,8] have mechanism to avoid or remove redundant patches.
De-duplication of elements is an important part of all patch discovery algorithms, but unfortunately previous algorithms use hand-tuned thresholds for de-duplication which makes algorithms difficult to compare. So instead, our work uses a greedy selection criterion that is the same for all algorithms; this selection process is designed to optimize for the coverage metric plotted in the curve, and represents the “best” de-duplication for this metric. So, while we could have used the de-duplication schemes to measure performance for these previous works, it would have actually resulted in significantly lower performance for these algorithms.

R10: Isn't "double-counting" actually necessary …...Or does "double-counting" refer to something else?
Yes, the intended meaning of “double counting” in this section is that a single patch may be a member of two or more different element clusters. This indicates that the two elements are representing the same thing, which is wasteful from a computational standpoint. We will clarify this.

R11: “I have seen [5] and [8] in CVPR 2013 and it seems to me that they both have LDA retraining”
We have asked the authors of both papers for clarification, and it seems that [8] does use LDA retraining, so it is indeed more similar to the “LDA retrained” baseline. [5] uses LDA for initialization only, and then switches to SVM retraining. We will clarify this.

R10: The algorithm with competition bears resemblance to k-means.
We agree that this resemblance exists, especially insofar as both k-means and the competition portion of our algorithm try to prevent data points from becoming members of multiple clusters (though perhaps EM for Gaussian mixture models is even more relevant, due to the soft cluster membership). We will mention this.

R10: it is hard to get any insight as to why the proposed method outperforms....Is this due to the clustering objective or to other implementation details described in lines 406-422?
While we have a number of implementation details, they are all fairly standard and mostly in common with previous work, e.g. [16].